# Who is absent and why? Factors affecting doctor absenteeism in Bangladesh

**Mir Raihanul Islam**[1], **Blake Angell**[2]*, **Nahitun Naher**[3], **Bushra Zarin Islam**[4], **Mushtaq Husain Khan**[5], **Martin McKee**[6], **Eleanor Hutchinson**[7], **Dina Balabanova**[7], **Syed Masud Ahmed**[3]

**1** Poverty, Gender and Inclusion Division, International Food Policy Research Institute, Dhaka, Bangladesh, **2** The George Institute for Global Health, University of New South Wales Sydney, Sydney, Australia, **3** BRAC James P Grant School of Public Health, BRAC University, Dhaka, Bangladesh, **4** School of Public Health and Health Sciences, University of Massachusetts Amherst, Amherst, Massachusetts, United States of America, **5** School of Oriental and African Studies (SOAS), University of London, London, England, **6** Department of Health Services Research and Policy, London School of Hygiene & Tropical Medicine, London, United Kingdom, **7** Department of Global Health and Development, Faculty of Public Health and Policy, London School of Hygiene and Tropical Medicine, London, United Kingdom

* bangell@georgeinstitute.org.au

## Abstract

Absenteeism by doctors in public healthcare facilities in rural Bangladesh is a form of chronic rule-breaking and is recognised as a critical problem by the government. We explored the factors underlying this phenomenon from doctors' perspectives. We conducted a facility-based cross-sectional survey in four government hospitals in Dhaka, Bangladesh. Junior doctors with experience in rural postings were interviewed to collect data on socio-demographic characteristics, work and living experience at the rural facilities, and associations with professional and social networks. Multiple logistic regression was used to determine the factors associated with rural retention. Of 308 respondents, 74% reported having served each term of their rural postings without interruptions. The main reasons for absenteeism reported by those who interrupted rural postings were formal training opportunities (65%), family commitments (41%), and a miscellaneous group of others (17%). Almost half of the respondents reported unmanageable workloads. Most (96%) faced challenges in their last rural posting, such as physically unsafe environments (70%), verbally abusive behaviour by patients/caregivers (67%) and absenteeism by colleagues that impacted them (48%). Respondents who did not serve their entire rural posting were less likely to report an unmanageable workload than respondents who did (AOR 0.39, 95% CI 0.22–0.70). Respondents with connections to influential people in the local community had a 2.4 times higher chance of serving in rural facilities without interruption than others (AOR 2.40, 95% CI 1.26–4.57). Our findings demonstrate that absenteeism is not universal and depends upon doctors' socio-political networks. Policy interventions rarely target unsupportive or threatening behaviour by caregivers and community members, a pivotal disincentive to doctors' willingness to work in underserved rural areas. Policy responses must promote opportunities for doctors with weak networks who are willing to attend work with appropriate support.

**Data Availability Statement:** We are happy to make available all data relevant to the analysis in the manuscript. As envisaged in our data management plan, we cannot make the raw data

fully open access due to its highly sensitive nature. But we are making the data available to our collaborators' institutions for further analysis, for example, for PhD theses. Others looking to access and use the data can send details of their request to Duncan Edwards, SOAS-ACE, at d.edwards@soas.ac.uk.

**Funding:** This publication is an output of the SOAS Anti-Corruption Evidence (ACE) research consortium funded by UK aid from the UK Government (Contract P07073 awarded to MHK). The views presented are those of the author(s) and do not necessarily reflect the UK government's official policies or the views of SOAS-ACE or other partner organisations. The funders had no role in study design, data collection and analysis, decision to publish, or preparation of the manuscript. For more information on SOAS-ACE, visit www.ace.soas.ac.uk.

**Competing interests:** The authors have declared that no competing interests exist.

## 1. Introduction

Health workforce shortages and doctor absenteeism are well-established facts in low- and middle-income countries like Bangladesh, jeopardising the population's fundamental right to health, especially in rural areas [1,2]. For the purposes of this paper that focuses on rural facilities, absenteeism is defined as leaving a rural posting early, for any reason, reduces the effectiveness of healthcare investments, compromises quality of services, and disproportionately affects vulnerable communities [3]. It increases the burden on present health workers, decreasing their motivation and performance [4,5]. Absenteeism also compounds existing system challenges relating to health workers, including shortages, maldistribution and poor retention, especially in rural and remote areas [6].

Absenteeism has been a long-standing problem facing the Bangladeshi health system, particularly among doctors [7,8]. The Ministry of Health and Family Welfare estimates the average rate of absence of doctors is 58.7% [9,10], while other estimates range from 35% to as high as 74% in single-doctor facilities [7,11]. Doctor absenteeism in Bangladesh disproportionately affects rural communities that already experience reduced access to care relative to their urban counterparts. There are only 1.1 doctors per 10 000 population in rural areas in Bangladesh, compared to 18.2 per 10 000 in urban areas [12]. Thirty-five per cent of doctors and 30% of nurses serve 15% of the total population living in four major cities of Bangladesh, including Dhaka, Chattogram, Rajshahi, and Khulna. In contrast, less than 20% of the health workforce serves over 70% of people living in rural areas [13]. The vacancy rates for doctor posts are also significantly higher in rural health facilities compared to urban areas [14] with 58.5% of doctor-level positions unfulfilled at the rural health complexes [15] compared to an average of 39.4% throughout Bangladesh [16].

Although doctors' absenteeism has long been recognised by the Bangladesh health system and multi-lateral and donor agencies, regulatory approaches such as installing fingerprint scanners to verify attendance and launching disciplinary action against absent doctors to overcome the problem have been unsuccessful [17,18]. This reflects a lack of evidence for top-down accountability and governance measures in overcoming absenteeism and other forms of health system corruption globally [19]. Experts have increasingly argued that these measures have been ineffective because they do not address the key incentives driving the behaviour of doctors nor account for the different contexts they face [1,2,20,21]. Instead, we argue that interventions targeting the critical drivers of doctor absenteeism offer the greatest promise for reducing absenteeism across the system. Bangladeshi doctors have strong but varied preferences for interventions to overcome absenteeism. We previously generated evidence suggesting that interventions considering the doctors' perspectives themselves could substantially reduce absenteeism [1]. To develop such interventions, it is vital to understand the critical drivers of absenteeism by doctors and the characteristics of those more likely to be absent. There is evidence that multiple factors influence doctors' availability in rural and remote public health facilities [22]. In qualitative research, doctors in Bangladesh reported rural facilities as more difficult and sometimes dangerous workplaces [22]. Heavy workloads (often increased by the absenteeism of others) and poor relationships of doctors with local communities made work stressful and sometimes unmanageable, impacting their other obligations, such as studying for exams and family commitments. Depending on their level of political connections, doctors reported varying degrees of absenteeism: those with influence, power and access to networks could be absent from their duty places for long periods, often without consequence [23].

There are limited empirical data investigating the prevalence of any of these factors or their association with doctor absences. We sought to help fill this gap by surveying Bangladeshi

doctors with recent experience working in rural facilities to investigate their characteristics and experiences and how they were associated with absenteeism.

## 2. Materials and methods

A survey was conducted in 2019 with doctors who are currently working at four tertiary hospitals in Dhaka and have experience working at rural facilities within the past ten years.

### 2.1 Survey design

The survey was developed based on the findings of qualitative work with a purposively selected sample of 30 doctors who work or have recently worked in facilities in rural Bangladesh [24]. The interviews explored their perceptions of what drives absenteeism and their views on potential solutions. Emerging themes informed the development of this survey based on the most crucial factors for rural absenteeism. The questionnaire was designed to elicit socio-demographic characteristics, educational background, professional affiliation, job-related information and rural placement information. On top of these categories, we asked questions on the experience and challenges faced by doctors in their last rural post, their self-assessed financial situation, and self-reported connections to powerful or influential people, as these factors were reported to influence absenteeism in the qualitative work. All doctors in Bangladesh must serve two years in a rural facility following their recruitment in the public sector, and we defined absenteeism as respondents reporting that they left this post early. The questionnaire was initially developed in English and later translated into Bangla for a better understanding by the field investigators.

### 2.2 Approval for data collection

The LSHTM Ethics Committee approved the project (Ref. 16 248) and the Institutional Review Board of BRAC James P Grant School of Public Health (ref. 2017–012). Formal approval for data collection was also obtained from the Directorate General of Health Services (DGHS), Ministry of Health and Family Welfare (MoHFW) and respective facility authorities. All the selected facilities issued permission letters highlighting the departments where data may be collected as needed. Detailed information about ethical, cultural, and scientific considerations related to inclusivity in global research can be found in S1 Checklist.

### 2.3 Sample

The survey was conducted in 2019 amongst doctors working at four tertiary hospitals with experience working in rural facilities. The facilities were chosen strategically to maximise recruitment, given the budget available for data collection. The target sample size for the study was calculated using the following formula:

$$n = \frac{Z_{1-\frac{\alpha}{2}}^2 \times p \times (1-p)}{d^2},$$

where n is the target sample size, $Z$ is the standard normal distribution (1.96), $\alpha$ is the level of significance (0.05), p is the population (assumed to be 0.50 since this would provide the maximum sample size) [25]. This gave a sample size of 384. The sampling strategy of this study was in line with the target sample size for the Discrete Choice Experiment survey. While there was no precise power calculation for the DCEs, a target sample size of this study was 300 [1]. After the survey, 308 doctors were included in the final analysis. Doctors were enrolled in the survey following convenience sampling.

### 2.4 Piloting

The questionnaire was piloted in two rounds among 15 doctors from two tertiary hospitals in Dhaka to ensure its acceptability, appropriateness, and understandability. This was followed by short interviews where respondents were asked for feedback on the questions and process. Respondents expressed concern over the questionnaire length through this process, so the tool was further revised, shortened, and finalised based on this feedback.

### 2.5 Data collection

Trained data collectors, under the guidance of the researchers, administered the survey (alongside a related discrete choice experiment) [1]. Based on pre-test experience, a varied approach was applied to collect data. This included group briefing sessions for all the duty doctors in particular departments where team members oriented the respondents on the study's background, aims, and objectives. The team used a one-to-one approach in other departments to brief individual respondents and collect data. Finally, in some situations, the team distributed the tool among eligible respondents and collected it on the following days as the doctors did not have time to fill it out. Data were collected in all facilities at a time convenient to the respondents. Prior communications were also made to eligible respondents to fix the date, time, and venue for data collection. Junior to mid-level doctors who had worked in rural facilities in the last ten years and above were approached. In all the hospitals, a Senior Research Associate monitored and supervised data collection, and overall activities were coordinated and monitored by the study's principal investigator (PI) and co-PI.

### 2.6 Operational definitions

As noted in the introduction, 'absenteeism' was defined as the doctors leaving their rural posting early for any reason, whereas 'uninterrupted service' means that doctors are fulfilling their rural postings without taking any breaks or leaving the assigned area without obtaining official permission. We have defined 'staff' as doctors, nurses, and those in other supporting roles in the hospital. 'Political connections' refer to any affiliations, associations, or relationships that individuals may have with local politicians or higher-level government officials. These connections could encompass personal relationships, affiliations with political parties or interest groups, or any form of interaction or influence that individuals have with those in positions of political power.

### 2.7 Analysis

A descriptive analysis of the demographic characteristics of the respondents was performed. Chi-squared tests were used to compare the prevalence of performing rural posting uninterruptedly within different categories of the variables at a 5% significance level. Crude and adjusted logistic regression models were run to assess the potential factors of absenteeism. The crude odds ratio (COR) and the adjusted odds ratio (AOR) were calculated using regression models. The final regression model included variables with p-values less than 0.25 in the crude model. This selection strategy was followed to simplify the model and avoid multicollinearity [26,27]. The adequacy and goodness-of-fit of the final regression model were diagnosed with the link test and Hosemer-lameshow's goodness-of-fit test. Results from the diagnostic tests reveal no misspecification in the model, and the regression model fitted the data. The variance inflation factor (VIF) was tested to detect multicollinearity in the independent variables. All the covariates showed VIF values less than three, so there was no indication of multicollinearity [28].

The crude odds ratio, adjusted odds ratio, and 95% CI were estimated at 5% significance level. All statistical analyses were completed using Stata (Version 13.0) [29].

## 3 Results

### 3.1 General characteristics

Three hundred and eight doctors with experience of rural posting completed the survey. Table 1 outlines their general characteristics: the majority of the respondents were aged 31–35 years (58%), male (54%), married (92%) and financially solvent (95%). Thirty-one per cent of the respondents were MBBS graduates, while the rest were enrolled or completed post-graduation. Currently, 47% of the respondents work as medical officers at the assigned hospitals.

**Table 1. Sociodemographic profile of the respondents.**

| Characteristics | Percentage (%) | No. of observations (n) |
|---|---|---|
| **Age** | | |
| 30 or less than 30 | 18.5 | 57 |
| 31–35 | 57.8 | 178 |
| 36–40 | 13.6 | 42 |
| 41 and above | 10.1 | 31 |
| **Sex** | | |
| Male | 53.6 | 165 |
| Female | 46.4 | 143 |
| **Religious affiliation** | | |
| Islam | 87.3 | 268 |
| Hinduism | 12.7 | 39 |
| **Marital status** | | |
| Married | 91.9 | 283 |
| Single and others | 8.1 | 25 |
| **Perceived financial situation in the last year** | | |
| Not deficit | 54.9 | 169 |
| Break-even | 40.3 | 124 |
| Deficit | 4.9 | 15 |
| **Completed MBBS from** | | |
| Public medical college | 90.9 | 279 |
| Private medical college | 9.1 | 28 |
| **Highest degree obtained** | | |
| MBBS | 30.5 | 94 |
| Enrolled in post-graduation | 55.5 | 171 |
| Completed post-graduation | 14.0 | 43 |
| **Current designation** | | |
| Medical officer | 47.4 | 145 |
| Registrar/ Asst. registrar[a] | 19.9 | 61 |
| Officer on special duty (OSD)[b] | 17.7 | 54 |
| Resident medical officer | 7.8 | 24 |
| Consultants and others | 7.2 | 22 |

A registrar is a mid-senior level position of physicians with a clinical specialty.

[b] An OSD is an officer with no specific task assigned to him.

## 3.2 Experiences at the rural facilities

Table 2 outlines the experiences of doctors in their rural postings. Around 26% said that they did not serve their mandatory two-year rural posting uninterruptedly, and the main reasons for doing so were availing training opportunities (65%) and family commitments (41%). Almost all respondents (96%) stated that they faced challenges in rural postings, such as physically unsafe environment (73%), verbal abuse (70%) and staff being absent (50%). To mitigate these challenges, respondents had to seek help from family (40%), locally influential people (39%), colleagues and neighbours (35%) and others. Seventy-five per cent of the respondents reported that there was a shortage of doctors in the facilities. The underlying reasons included doctors not being posted against sanctioned posts (63%) and doctors leaving their posts to pursue higher education (45%). Forty-eight percent of the respondents complained of a workload that was too high, which sometimes became unmanageable. At the same time, around 30% of the respondents said they were not involved in private practice during their rural postings. Almost two-thirds (63%) of the respondents did not stay in the residence provided by the authority most of the time, and over three-quarters (76%) of respondents' families did not stay in the residence provided. The primary reasons for not staying in residence provided by the administration were lack of security (77%), lack of basic amenities (72%), and a lack of convenient transport services (40%).

A significantly greater proportion of doctors (82%) who served their rural post uninterruptedly reported that the workload in the rural facilities was not manageable, and around 67% of the doctors who served their rural post uninterruptedly thought the workload was manageable (p = 0.004) (Table 3). Also, the respondents' ability to seek help from influential people in the local community if faced with any local problem was significantly associated with their uninterrupted rural posting (p = 0.003).

Table 4 shows the bivariate relationship between factors favourable to career progression and serving rural posting uninterruptedly. A significantly higher proportion of doctors who completed rural postings revealed that personal networks were helpful in the career progression of doctors (84% vs 71%, p = 0.034).

## 3.3 Factors associated with absenteeism

Logistic regression models were run to predict factors associated with uninterrupted services during rural posting (Table 5). Crude and adjusted models were considered to identify the association between networking and rural experience-related factors and continuous services during a rural posting. In the crude model, it was found that respondents aged 41 and above had a 64% lesser chance of serving the rural posting than respondents aged 30 or less. (AOR 0.36, 95% CI 0.14–0.93). Personal networks in career progression played a substantial role in serving rural facilities uninterruptedly compared to those who lack these connections (AOR 2.12, 95% CI 1.05–4.28). Also, respondents who had a connection with influential people in the local community had a 2.4 times higher chance of serving in rural facilities uninterruptedly compared to the respondents who didn't have any such connection (AOR 2.4, 95% CI 1.05–4.28). Respondents who served the entire posting time in the rural facility found the workload 54% less manageable than the respondents who didn't serve the entire posting period in a rural facility (AOR 0.46, 95% CI 0.26–0.80).

In the adjusted model, all factors were considered to identify their association with uninterrupted continuation of service. It is evident from the findings that connection with local influential figures and workload were the only two significant factors. Respondents having connections with influential people in the local community had a 2.6 times higher chance of serving in rural facilities uninterruptedly than the respondents who didn't have such

**Table 2. Experience of doctors at the rural facilities.**

| | Percentage (%) | No. of observations (n) |
|---|---|---|
| **Served each period of rural posting uninterruptedly** | | |
| Yes | 74.3 | 225 |
| No | 25.7 | 78 |
| **If not, then the reason for leaving the posting early (Multiple responses)** | | |
| Training opportunities offered by the authority | 65.4 | 51 |
| Family reasons | 41.0 | 32 |
| Post-graduation and others | 16.7 | 13 |
| **Challenges faced in the last rural facility** | | |
| Yes | 96.4 | 297 |
| No | 3.6 | 11 |
| **If yes, then what types of challenges faced in the last rural posting during service delivery (multiple responses allowed)** | | |
| Felt physically unsafe | 73.1 | 215 |
| Verbal abuse by caregivers/community members | 69.7 | 205 |
| Staff being absent in working hours | 50.3 | 148 |
| No co-operation from colleagues | 27.2 | 80 |
| Theft | 26.2 | 77 |
| Physical abuse by caregivers/community members | 15.3 | 45 |
| Others | 9.9 | 28 |
| **Helps sought for coping in crisis (multiple responses allowed)** | | |
| Family | 39.8 | 121 |
| Locally influential people | 38.8 | 118 |
| Social networks (Colleagues) | 34.5 | 105 |
| Friend | 32.6 | 99 |
| Others | 2.0 | 6 |
| None of the above | 12.5 | 38 |
| I didn't need any help | 2.6 | 8 |
| **Doctor shortage in last rural posting** | | |
| Yes | 74.5 | 228 |
| No | 25.5 | 78 |
| **If yes, then the reason for the doctor shortage in the last rural facility (multiple responses allowed)** | | |
| Doctors not posted in sanctioned post | 63.0 | 143 |
| Doctor left the post to join post-graduation | 44.5 | 101 |
| Doctors absent without official reason | 7.9 | 18 |
| Inadequate posts and others | 11.4 | 26 |
| **Workload in the rural facility** | | |
| Manageable, as expected | 14.7 | 44 |
| Mostly manageable, occasionally too much | 37.5 | 112 |
| Usually too much/ not manageable | 47.8 | 143 |
| **Private practice during rural posting** | | |
| Yes | 29.7 | 91 |
| No | 70.3 | 215 |
| **Lived in the residence provided by the health facility** | | |
| Yes, most of the time | 37.0 | 114 |
| Yes, sometimes | 16.9 | 52 |
| Never | 24.0 | 74 |
| Not applicable-accommodation was not available | 14.0 | 43 |
| Had own accommodation | 8.1 | 25 |

*(Continued)*

**Table 2.** (Continued)

| | Percentage (%) | No. of observations (n) |
|---|---|---|
| **Family stayed at the rural posting** | | |
| Yes | 24.1 | 40 |
| No | 75.9 | 126 |
| **Reason for not staying in health facility residence (multiple responses)** | | |
| Lack of security (guard, gate etc.) | 76.7 | 155 |
| Lack of basic amenities | 72.3 | 146 |
| Lack of convenient transport | 39.6 | 80 |
| Family reasons | 39.1 | 79 |
| Lack of quality education for children | 33.2 | 67 |
| To avoid being disturbed by patient(s) during off-duty hours | 21.8 | 44 |

connections (AOR 2.64, 95% CI 1.37–5.08). Then again, respondents who served the entire posting period in the rural facility found the workload 59% less manageable compared to the respondents who didn't serve the entire posting period in a rural facility (AOR 0.41, 95% CI 0.22–0.73).

## 4 Discussion and conclusions

About one-quarter of the cohort of doctors in this study with recent experience working in rural Bangladesh reported leaving their rural posting early. Doctors reported facing significant challenges working in rural areas, including feeling physically unsafe, experiencing verbal and physical abuse, staff shortages, non-cooperative colleagues, theft, and poor-quality housing. Doctors who reported being well-supported by the community through access to influential people were more than twice as likely as others to complete their rural posting. While some previous literature has highlighted these doctors' challenging conditions, little quantitative data has been published on the type and extent of these challenges. These results provide a compelling (albeit partial) explanation for the historical difficulties observed in overcoming absenteeism in these facilities, as well as the broader issues of attracting and retaining doctors to serve rural areas in Bangladesh.

As noted, we found that doctors who felt they were well-supported by the local community through access to influential people were over twice as likely to complete their full term of service in the place of rural posing. Interestingly, those who reported an unmanageable workload were found to be less likely to report leaving early. While this may initially seem counterintuitive, the finding aligns with other emerging literature highlighting the importance of social and political networks for doctors in securing preferred urban postings; those without such networks had no other alternative to continue despite heavy workload. Taken together, our findings highlight the importance of contextual factors in determining rural absenteeism and provide further evidence as to why traditional top-down approaches to doctor absenteeism have so far been ineffective.

This study also heard reports that doctors in rural areas must leave their posts frequently to complete official training (short or long-term) or higher studies sponsored by the authorities. Such schemes operate against efforts to secure a stable health workforce for the rural population. It can take a long time for lost doctors to be replaced, given complex administrative procedures and the shortage of doctors to fill the newly created vacancies [30,31]. It also doubly penalises doctors in rural areas as they are more likely to be overworked and less likely to access such training opportunities. This experience mirrors that in other nations [32–34].

**Table 3. Work experience and challenges in rural facility during posting faced by the respondents who served rural posting uninterruptedly.**

| Experience in a rural facility | served rural posting uninterruptedly | Didn't serve rural posting uninterruptedly | P value[a] |
|---|---|---|---|
| | % (n) | % (n) | |
| Workload in rural facility | | | |
| Manageable | 66.9 (103) | 33.1 (51) | 0.004* |
| Not manageable | 81.6 (115) | 18.4 (26) | |
| Private practice during rural posting | | | |
| No | 75.4 (159) | 24.6 (26) | 0.442 |
| Yes | 71.1 (64) | 28.9 (52) | |
| Lived in the residence provided by the health facility | | | |
| No | 78.3 (108) | 21.7 (30) | 0.145 |
| Yes | 70.9 (117) | 29.1 (48) | |
| **Challenges in rural facility** | | | |
| Felt physically unsafe environment | | | |
| No | 77.8 (70) | 22.2 (20) | 0.362 |
| Yes | 72.8 (155) | 27.2 (58) | |
| Experienced physically abusive behaviour | | | |
| No | 73.8 (191) | 26.2 (68) | 0.621 |
| Yes | 77.3 (34) | 22.7 (10) | |
| Experienced verbally abusive behaviour | | | |
| No | 68.6 (70) | 31.4 (32) | 0.110 |
| Yes | 77.1 (155) | 22.9 (46) | |
| Staffs were absent in working hours | | | |
| No | 71.3 (112) | 28.7 (45) | 0.228 |
| Yes | 77.4 (113) | 22.6 (33) | |
| Seek help from family if you faced any problem locally | | | |
| No | 76.1 (140) | 23.9 (44) | 0.365 |
| Yes | 71.4 (85) | 28.6 (34) | |
| Seek help from friends if you faced any problems locally | | | |
| No | 75.7 (156) | 24.3 (50) | 0.393 |
| Yes | 71.1 (69) | 28.9 (28) | |
| Seek help from social links if you faced any problems locally | | | |
| No | 75.9 (151) | 24.1 (48) | 0.372 |
| Yes | 71.2 (74) | 28.8 (30) | |
| Seek help from influential people in the local community if you faced any problem locally | | | |
| No | 68.3 (127) | 31.7 (59) | 0.003* |
| Yes | 83.8 (98) | 16.2 (19) | |
| Doctor shortage in last rural posting | | | |
| Yes | 73.5 (166) | 26.5 (60) | 0.662 |
| No | 76.0 (57) | 24.0 (18) | |
| If yes, then the reason for the doctor shortage in the last rural facility (multiple responses) | | | |
| Doctors not posted in sanctioned post | 76.6 (108) | 23.4 (33) | 0.385 |
| Doctor left the post to join post-graduation | 72.3 (73) | 27.7 (28) | 0.577 |
| Doctors absent without official reason | 94.4 (17) | 5.6 (1) | 0.050 |

[a]P values are obtained from the chi-square test.

* Significant at a 5% level of significance.

**Table 4. Factors favourable to career progression.**

| Favourable factors | served rural posting uninterruptedly | Didn't serve rural posting uninterruptedly | P value[a] |
|---|---|---|---|
| | % (n) | % (n) | |
| Post-graduation helped in career progression | | | |
| No | 79.0 (15) | 21.0 (4) | 0.629 |
| Yes | 73.9 (210) | 26.1 (74) | |
| Promotional exams helped in career progression | | | |
| No | 73.4 (116) | 26.6 (42) | 0.727 |
| Yes | 75.2 (109) | 24.8 (36) | |
| Money helped in career progression | | | |
| No | 73.4 (157) | 26.6 (57) | 0.581 |
| Yes | 76.4 (68) | 23.6 (21) | |
| Personal network helped in career progression | | | |
| No | 71.4 (167) | 28.6 (67) | 0.034* |
| Yes | 84.1 (58) | 15.9 (11) | |
| Family access to influential people | | | |
| No | 76.0 (73) | 24.0 (23) | 0.629 |
| Yes | 73.4 (152) | 26.6 (55) | |

[a] P values are obtained from the chi-square test.

* Significant at a 5% level of significance.

Policies on training need to be carefully calibrated such that they do not create a situation whereby doctors are incentivised not to complete postings in rural communities. This view is supported by results from a discrete choice experiment in this population that points to the value of prioritising doctors with a good attendance record through the provision of 'bonus points' that enable preferential placement in higher education or training [1].

There is a common belief that the doctors remaining in the rural postings are compensated by flourishing private practice, an opportunity to be with the family, or a residence near the workplace provided by the authority [22,35]. However, this study did not find a significant relationship between private practice (almost 30% of our sample) and completing the rural posting period. Ultimately, our work suggests that the experience junior doctors have when posted to a rural facility is likely to be challenging but will vary widely depending on how safe and supported they feel. Addressing these challenges should be a priority, and these findings have important implications for policies seeking to overcome doctor absenteeism in Bangladesh by providing a unique insight into doctors' perspectives on a problem that has largely been neglected, aside from a few related studies [1,30,36]. While attention has traditionally focused on increased monitoring of doctors and enforcement of regulations (for example, punishing rule breakers), our results build on existing literature and suggest that it may be more effective to focus attention and scarce health system resources on improving the working conditions facing doctors who wish to be present at work. Forging strong linkages with local communities so that providers receive support and protection from violence appears to be a promising channel for further investment [10,37,38]. Further, allowing for flexibility in circumstances where doctors have opportunities to attend training courses to improve their skills without being penalised will likely be highly valued by doctors. It could improve their retention and motivation to serve in rural communities [39].

Our research was subject to several limitations. Resource constraints limited our sample to doctors working in four large urban hospitals with recent experience working at rural facilities.

**Table 5. Factors associated with uninterrupted service during rural posting.**

| Factors | Crude model | | | Adjusted model | | |
|---|---|---|---|---|---|---|
| | COR | 95% CI | p-value | AOR | 95% CI | p-value |
| Age | | | | | | |
| 30 or less than 30 | 1 | | Reference | 1 | | Reference |
| 31–35 | 0.79 | 0.38–1.62 | 0.518 | 0.80 | 0.37–1.74 | 0.582 |
| 36–40 | 1.16 | 0.43–3.15 | 0.772 | 1.58 | 0.54–4.62 | 0.401 |
| 41 and above | 0.36 | 0.14–0.93 | 0.036 | 0.44 | 0.16–1.23 | 0.119 |
| Gender | | | | | | |
| Male | 1 | | Reference | 1 | | Reference |
| Female | 0.71 | 0.42–1.19 | 0.192 | 0.76 | 0.42–1.41 | 0.389 |
| Preferred career path | | | | | | |
| Health service delivery | 1 | | Reference | 1 | | Reference |
| Health professional education & training (Academic) | 0.54 | 0.15–1.91 | 0.336 | 0.53 | 0.12–2.31 | 0.399 |
| Health administration | 0.63 | 0.34–1.18 | 0.149 | 0.57 | 0.28–1.15 | 0.117 |
| Personal network helped in career progression | | | | | | |
| No | 1 | | Reference | 1 | | Reference |
| Yes | 2.12 | 1.05–4.28 | 0.037 | 1.18 | 0.85–3.88 | 0.124 |
| **Work experience from rural facility** | | | | | | |
| Seek help from influential people in the local community | | | | | | |
| No | 1 | | Reference | 1 | | Reference |
| Yes | 2.40 | 1.34–4.28 | 0.003 | 2.64 | 1.37–5.08 | 0.004 |
| Experienced verbally abusive behaviour | | | | | | |
| No | 1 | | Reference | 1 | | Reference |
| Yes | 1.21 | 0.57–2.58 | 0.621 | 0.87 | 0.37–2.08 | 0.763 |
| Staffs were absent in working hours | | | | | | |
| No | 1 | | Reference | 1 | | Reference |
| Yes | 1.38 | 0.82–2.31 | 0.229 | 1.24 | 0.69–2.24 | 0.469 |
| Workload in rural facility | | | | | | |
| Not manageable | 1 | | Reference | 1 | | Reference |
| Manageable | 0.46 | 0.27–0.79 | 0.005 | 0.41 | 0.22–0.73 | 0.003 |
| Lived in a residence provided by the health facility | | | | | | |
| No | 1 | | Reference | 1 | | Reference |
| Yes | 0.68 | 0.40–1.15 | 0.146 | 0.61 | 0.34–1.08 | 0.091 |

Note: Results are based on simple and multiple logistic regression, COR: Crude Odds Ratio, AOR: Adjusted Odds Ratio, 95% CI: 95% Confidence interval.

Thus, they may not be generalisable to the broader medical population servicing rural communities. Given the sensitivity of absenteeism in Bangladesh, respondents may not have been comfortable being open when responding to some of the questions. We attempted to minimise the impact of such bias through an extensive survey development process, including qualitative work and piloting with the target population. Despite these steps, recruitment was challenging, and we fell short of our target sample size. Nonetheless, the significance and fit of our models suggest that we have identified significant relationships between our variables of interest and the absenteeism of doctors at rural postings in Bangladesh.

Despite these limitations, this work represents a unique contribution to the literature by providing empirical data from the doctors' perspective in Bangladesh. This perspective is surprisingly overlooked in debates about doctor absenteeism. Our findings build on recent developments in the field, suggesting that it may be more pragmatic and cost-effective to focus on

meeting the needs of doctors who can be induced to attend work rather than monitoring and punishing offenders.

## Supporting information

**S1 Checklist. Inclusivity in global research.**
(DOCX)

## Author Contributions

**Conceptualization:** Mushtaq Husain Khan, Dina Balabanova.

**Data curation:** Mir Raihanul Islam.

**Formal analysis:** Mir Raihanul Islam, Blake Angell.

**Funding acquisition:** Mushtaq Husain Khan, Dina Balabanova.

**Investigation:** Mir Raihanul Islam, Blake Angell, Nahitun Naher.

**Methodology:** Mir Raihanul Islam, Blake Angell, Martin McKee, Eleanor Hutchinson, Dina Balabanova, Syed Masud Ahmed.

**Project administration:** Nahitun Naher, Bushra Zarin Islam.

**Supervision:** Nahitun Naher, Mushtaq Husain Khan, Dina Balabanova, Syed Masud Ahmed.

**Writing – original draft:** Mir Raihanul Islam.

**Writing – review & editing:** Mir Raihanul Islam, Blake Angell, Nahitun Naher, Bushra Zarin Islam, Mushtaq Husain Khan, Martin McKee, Eleanor Hutchinson, Dina Balabanova, Syed Masud Ahmed.

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
