## [Decision Letter · Decision Letter 0]

31 Aug 2023

PGPH-D-23-01040

Who is absent and why? Factors affecting doctor absenteeism in Bangladesh

Dear Dr. Angell,

Thank you for submitting your manuscript to PLOS Global Public Health. After careful consideration, we feel that it has merit but does not fully meet PLOS Global Public Health’s publication criteria as it currently stands. Therefore, we invite you to submit a revised version of the manuscript that addresses the points raised during the review process.

We look forward to receiving your revised manuscript.

Kind regards,

Veena Sriram

Academic Editor

Journal Requirements:

2. Please send a completed 'Competing Interests' statement, including any COIs declared by your co-authors. If you have no competing interests to declare, please state "The authors have declared that no competing interests exist". Otherwise please declare all competing interests beginning with twhe statement "I have read the journal's policy and the authors of this manuscript have the following competing interests:"

3. Please amend your detailed Financial Disclosure statement. This is published with the article. It must therefore be completed in full sentences and contain the exact wording you wish to be published.

4. Please provide separate figure files in .tif or .eps format only and remove any figures embedded in your manuscript file. Please also ensure all files are under our size limit of 10MB.

5. We have noticed that you have a list of Supporting Information legends in your manuscript. However, there are no corresponding files uploaded to the submission. Please upload them as separate files with the item type 'Supporting Information'. 

6. Tables should not be uploaded as individual files. Please remove these files and include the Tables in your manuscript file as editable, cell-based objects. For more information about how to format tables, see our guidelines:

https://journals.plos.org/globalpublichealth/s/tables

Additional Editor Comments (if provided):

Reviewers' comments:

Reviewer's Responses to Questions

**Comments to the Author**

1. Does this manuscript meet PLOS Global Public Health’s publication criteria? Is the manuscript technically sound, and do the data support the conclusions? The manuscript must describe methodologically and ethically rigorous research with conclusions that are appropriately drawn based on the data presented.

Reviewer #1: Partly

Reviewer #2: Partly

2. Has the statistical analysis been performed appropriately and rigorously?

Reviewer #1: Yes

Reviewer #2: I don't know

3. Have the authors made all data underlying the findings in their manuscript fully available (please refer to the Data Availability Statement at the start of the manuscript PDF file)?

Reviewer #1: Yes

Reviewer #2: Yes

4. Is the manuscript presented in an intelligible fashion and written in standard English?

Reviewer #1: No

Reviewer #2: Yes

5. Review Comments to the Author

Reviewer #1: This manuscript is based on research on an important topic and deserves publication. However, the authors need to address some issues, especially the discussion section requires significant revision.

Abstract:

“Respondents who did not serve….”: The meaning of the sentence is not clear. Please revise.

Introduction:

What does ‘regional communities’ mean in the second paragraph? Does it mean ‘rural communities’?

“An increasing body … all doctors equally”: The meaning of the sentence is not clear. Please revise.

“Instead, we argue … across the system”: Please revise to simplify the sentence.

“Depending on the degree … periods without sanction”: Please provide a reference. The same reference from Joarder et al. (2018) that was used before demonstrated that doctors' political affiliation contributed to their avoidance of the 2-year compulsory rural service.

Any reference will suffice, however.

Also, what does sanction mean in this sentence? Please revise for clarity.

Survey design:

‘or the impact of sanctions’: What does this mean?

“All doctors in Bangladesh … left this post early”: Not following their training; following their recruitment in the public sector. Please revise accordingly.

Analysis:

Please provide a reference to the software Stata (Version 13.0).

General characteristics:

“Table 1 outlines …: To align with the table, please report the age first, then gender, and other variables as per the sequence they are listed in the table.

Delete ‘(Table 5)’ from that sentence. It is incorrect.

“At present, 47% …”: No need to mention all categories in the text unless it imparts any important message. Please delete from 20% onwards.

Experiences at the rural facilities:

‘Almost all respondents’: Mention 96% within parenthesis.

‘transferred to sanctioned posts’: Transferred or posted. Please replace it with 'posted' if that is correct.

“About one third of the respondents…”: Instead of writing that 37% stayed in the residence, please write that more than three fourth of the respondent did not stay in the residences provided to them on campus. This sounds more intriguing.

‘lack of basic amenities (73%)’: Wrong. It is 72%, according to Table 2.

“A significantly greater proportion …”: Unclear sentence. Please revise.

“Interestingly, no significant … “: Meaning not clear. Also, the significance of this finding is not clear. Please delete it if not important.

Factors associated with absenteeism:

Mention Table 5, where these findings are documented.

“Three models were considered …”: I am not sure why the authors need three different models to convey the same message or findings, which they can easily do with one inclusive model (model iii). If there is no important reason, please revise this part (and the methods, too) to show only one model, including all variables.

“In model I, networking … “: In Table 5, this set of variables was termed as 'factors favorable for career progression'. Please use the correct term consistently and ensure the terms used are aligned between the text and the table.

AOR = 0.016: This is wrong. The AOR is 2.15. This is the p-value. Please revise.

“In the final … absenteeism.”: Absenteeism or uninterrupted service? These variables are different, so please clarify.

Discussion and conclusions:

Paragraph 1: This paragraph is redundant. The first sentence may be used in the background section, where the authors try to demonstrate the importance of the research topic. This is unnecessary in the discussion section as the importance or significance of the research is supposed to be established already.

‘facing doctors’: Instead, use ‘doctors face’.

“These include but … “: New findings are introduced in the discussion which are unsupported by data.

‘intelligent use’: Please avoid loaded phrases like this and report what you found exactly.

‘leave vacancies’: What is this?

“However, this study … to be significant”: Wonder why. Any explanation?

“While attention has … present at work”: Not sure how do the authors draw such a conclusion when their research does not include the former variables to compare?

‘paying attention to … provision of logistics’: How does the data support this?

‘pre-planning to ensure … transparent career path’: How is this found in the research?

“When these are combined … absenteeism of doctors”: How do the authors know that? Any evidence?

“Besides, WHO … at the earliest”: How is this related to the study?

The last paragraph: Not supported by the data presented in the manuscript.

Major concerns: Overall, the discussion and conclusion sections need to be overhauled.

In the discussion section:

1. State the study's major findings in the first paragraph in a direct, declarative and succinct manner. You may select two or, at best, three major findings from your study.

2. Write one paragraph or two for each major finding by explaining the meaning and importance of the findings.

3. Relate the findings to those of similar findings, preferably from Bangladesh or similar settings.

4. Mention the limitation and strengths of the study. This is somehow addressed already.

5. Provide your opinion about future research and/or policy implications, research/policy recommendations.

In the conclusion section or the last paragraph of the discussion section, give a take-home message, i.e., a couple of sentences summarizing your research and things you want the reader to remember, at the least from your article.

The current discussion section suffers from some serious flaws, such as over-interpretation of results, speculations, tangential issues, and, most importantly, the introduction of new issues that are not presented in the results and proposing conclusions or recommendations unsupported by data.

Reviewer #2: This paper is straightforward and clearly written. I have a few ovearching comments and several more minor comments that the authors might consider in order to make the analysis sharper.

First, the finding that respondents who did not serve the entire rural posting were less likely to report an unmangeable workload is somewhat counterintuitive. The authors could point this out (somone reading quickly might even understand the reverse) and explore why. For example, perhaps simply staying made one interpret the workload as more, or, there was a greater liklihood among those willing to take on a higher workload to also stay.

Second, the study explores both unsupportive or threatening behavior from caregivers and community members as distinct from political connections. What is the definition of political connections? How was this question asked? Is it to local politicians or higher up? Are these two constructs related conceptually in any way? (this isn't a measurement question so much as a question of your readers' ability to understand the findings).

The introduction defines absenteeism as unscheduled work absence, but the study seems to focus on leaving the post early (rather than disappearing to Dhaka for a month for training, for example). The exact defintiion used and to what extent this tracks with the broader literature should be explained.

The Introduction cites an estimate from the MoHFW regarding absenteeism. Is this 58.7% figure for any given day? (ie if I were to show up at every facility in the country on the same day, 58.7% of the HCWs would not be present). This is somewhat different from your leaving the post early definition.

The introduction refers to failed regulatory approaches the government has taken. It would be helpful to the reader to add a few words explaining what some of these have been.

The introduction also notes that interventions to overcome absenteeism do not impact all doctors equally. I understand what you mean, but perhaps I assume that your point is not about equality. Rather, it is about the extent to which these interventions meaningfully shift incentives in different contexts?

The section on survey deisgn should cite the qualitative work (presumably the 2015 study).

Section 3.1 of the results descibes the characteristics of the respondents. It might be helpful for your readers who don't know Bangladesh to define a registrar, and officer on special duty.

Section 3. 2: can you explain how social networks are distinct from locally influential people? (I can guess but better to be explicit!)

It would be helpful if Section 3.3 had a short table summarizing key findings.

Section 4 (discussion) refers to scarce logistics, pressure from local elites and political hoodlums for undue advantage... these were not explored in the results (or I missed them). How do these relate conceputally to attacks from the community, social networks, and political connections?

6. PLOS authors have the option to publish the peer review history of their article (what does this mean?). If published, this will include your full peer review and any attached files.

**Do you want your identity to be public for this peer review?** For information about this choice, including consent withdrawal, please see our Privacy Policy.

Reviewer #1: **Yes: **Taufique Joarder

Reviewer #2: No

---

## [Decision Letter · Decision Letter 1]

28 Dec 2023

PGPH-D-23-01040R1

Who is absent and why? Factors affecting doctor absenteeism in Bangladesh

Dear Dr. Angell,

Thank you for submitting your manuscript to PLOS Global Public Health. After careful consideration, we feel that it has merit but does not fully meet PLOS Global Public Health’s publication criteria as it currently stands. Therefore, we invite you to submit a revised version of the manuscript that addresses the points raised during the review process.

We look forward to receiving your revised manuscript.

Kind regards,

Veena Sriram

Academic Editor

Journal Requirements:

Additional Editor Comments (if provided):

Thank you for submitting this revision, and for your patience with the review process. The reviewer has provided a detailed review of this revision and finds that there continue to be several areas that require attention. You are invited to submit a revision that considers these comments carefully, particularly around methods and statistical analysis.

Reviewers' comments:

Reviewer's Responses to Questions

**Comments to the Author**

1. If the authors have adequately addressed your comments raised in a previous round of review and you feel that this manuscript is now acceptable for publication, you may indicate that here to bypass the “Comments to the Author” section, enter your conflict of interest statement in the “Confidential to Editor” section, and submit your "Accept" recommendation.

Reviewer #1: (No Response)

2. Does this manuscript meet PLOS Global Public Health’s publication criteria? Is the manuscript technically sound, and do the data support the conclusions? The manuscript must describe methodologically and ethically rigorous research with conclusions that are appropriately drawn based on the data presented.

Reviewer #1: Partly

3. Has the statistical analysis been performed appropriately and rigorously?

Reviewer #1: No

4. Have the authors made all data underlying the findings in their manuscript fully available (please refer to the Data Availability Statement at the start of the manuscript PDF file)?

Reviewer #1: No

5. Is the manuscript presented in an intelligible fashion and written in standard English?

Reviewer #1: No

6. Review Comments to the Author

Reviewer #1: The revised version of the manuscript shows improvement; however, there are still significant issues, particularly in the methods (sampling and analysis) and results (Tables 3, 4, 5 contain major errors, which are not acceptable at this stage of revision) sections. While the discussion section has improved from the earlier version, it does not fully align with the provided guidance, and some crucial explanations are still missing. Additionally, persistent formatting and syntax errors are present throughout the manuscript, which is not acceptable at this stage.

Abstract:

“Those who did not….”: The sentence reads confusing with a comma after "Those who did not."

Keywords:

Why is Network written with a capital N? There are inconsistent capitalization in other keywords too.

Introduction:

There incorrect formatting of in-text citations, notably, 1, 2, 3.

“It also increases…and remote areas”: Very long sentence. Break it into two.

“An increasing body … contexts they face.”: Please simplify this sentence.

“found that doctors … places to work.”: Incomplete sentence.

“Specifically, we defined … for any reason.”: Definition of absenteeism must be clarified much earlier; not as the last sentence of the background section. Shift this sentence to the place where you first introduced the term 'absenteeism.'

Overall comment: The writing style and sentence structure must be further tightened up. Please consider getting the manuscript thoroughly reviewed for language by a native English speaker.

Materials and methods:

Please add a section on sampling, including sample size calculation and sampling approach.

Analysis:

Please consult with a statistician to confirm if the application of a logistic regression analysis is appropriate in this context. This is particularly important as the authors did not clarify the sampling procedure.

Some important assumptions of logistic regression might have been violated:

1. Independence of Observations: The observations in the dataset are independent of each other. This assumption might have been violated if multiple doctors were surveyed from the same health facility.

2. Absence of multicollinearity: Nothing has been mentioned in the methods section if this has been checked.

3. No regression diagnostics have been reported in the manuscript.

Optional: One suggestion for improvement is to elaborate on the rationale for choosing the 0.25 threshold for variable inclusion in the final model. Providing a brief justification could enhance the transparency of this decision.

General characteristics:

“Around 26% said … family commitments (41%).”: What does ‘uninterrupted mean here? Few days of interruption, few weeks, months, not completing the full 2 years? Please provide an operation definition in the methods section.

“Almost all respondents … staff being absent (50%).”: What does staff mean here? Another physician, a nurse, any stuff? Please specify. Provide an operational definition in the methods section.

“To mitigate these challenges…”: there is a full stop after ‘neighbours’. Please correct it.

“At the same time … rural postings.”: This statement is not clear. They did not get a chance because they were not allowed to do private practice, or they did not find time for private practice due to other commitments such as higher education, family, etc.? Please write the results accordingly to what you actually asked in the questionnaire.

Table 2: “Served each period of rural posting uninterruptedly”: What does the 'each period of rural posting' mean? Why not just ask 'Served rural posting uninterruptedly?

Table 2, under ‘Workload in the rural facility’: What is the difference between the response categories ‘Usually too much to be manageable’ and Not manageable’? In my opinion, they don’t necessary impart any meaningful difference. If the authors agree, they should recode the response categories and re-run the analysis.

Table 3: Why do these percentages not add up to 100? I understand these are chi-squared tables, but this way of presenting the tables is confusing and does not convey a clear meaning. Please revisit this table.

Table 4: Same comment as Table 3. If you are trying to present your chi-squared test findings in a tabulated form, please use an appropriate table format. Perhaps you want to show only the significant association, i.e., the association between uninterrupted rural stay and personal network.

Table 5: This table has multiple mistakes:

1. Networking variables are not identified like career progression and work experience variables.

2. Network variables are wrongly identified as career progression variables.

3. Model 2 includes network variables too, as evidenced by the table, but the text does not say so. The text says that model 2 includes rural experience and socio-demographic variables.

Overall comment: I am unsure what additional value the complicated three-model analysis adds as opposed to a single full model. The significant variables in models 2 and 3 are the same. The only additional significant variable in model 1 is age, which may not be too intriguing. So, instead of three models, a single full model and an unadjusted model might be useful and easier to comprehend.

Discussion and conclusions:

In my earlier review, I suggested presenting the major findings directly and declaratively in the first paragraph of the discussion section. This has not been respected. I am repeating my suggestion again for improving the manuscript in its future iteration:

1. State the study's major findings in the first paragraph in a direct, declarative and succinct manner. You may select two or, at best, three major findings from your study.

2. Write one paragraph or two for each major finding by explaining the meaning and importance of the findings.

3. Relate the findings to those of similar findings, preferably from Bangladesh or similar settings.

4. Mention the limitation and strengths of the study in a separate paragraph.

5. Provide your opinion about future research and/or policy implications, research/policy recommendations. Do it in a separate paragraph instead of clubbing them with each paragraph.

6. In the conclusion section or the last paragraph of the discussion section, give a take-home message, i.e., a couple of sentences summarizing your research and things you want the reader to remember, at the least from your article.

“We found that … rural posting.”: This indicates that doctors with local support are more likely to stay in rural areas. This also means that the tendency of doctors to local stay is contingent on informal local support. This eventually means there is a lack of formal support system for the doctors to encourage and support them to stay there. These can be supported by other literature from Bangladesh and similar settings.

Also, delete the extra full stop from the sentence.

Also, this paragraph does not attempt to substantiate the discussions with findings from Bangladesh or similar contexts.

“It is also doubly … training opportunities.”: Instead of this, write, “It also doubly penalise doctors in the rural areas as they are both more likely to be overworked and the absence of some of their colleagues adds to the existing high workload.”

“However, this study … rural posting period.”: It is possible that their private practice is not in their area of rural posting but rather in a different, perhaps urban, area. This finding and the associated paragraph deserves more thoughtful explanations.

“Policies around training … education or training.”: This part of the paragraph is discrete from the previous arguments. Please create a separate paragraph.

“There are several limitations … target population.”: In my opinion, the most important limitation is the nonprobability sampling method of your study, although the authors did not mention anything about the sampling procedure.

7. PLOS authors have the option to publish the peer review history of their article (what does this mean?). If published, this will include your full peer review and any attached files.

**Do you want your identity to be public for this peer review?** For information about this choice, including consent withdrawal, please see our Privacy Policy.

Reviewer #1: **Yes: **Taufique Joarder

---

## [Decision Letter · Decision Letter 2]

29 Feb 2024

Who is absent and why? Factors affecting doctor absenteeism in Bangladesh

PGPH-D-23-01040R2

Dear Dr Angell,

We are pleased to inform you that your manuscript 'Who is absent and why? Factors affecting doctor absenteeism in Bangladesh' has been provisionally accepted for publication in PLOS Global Public Health.

Best regards,

Veena Sriram

Academic Editor

Reviewer Comments (if any, and for reference):

Reviewer's Responses to Questions

**Comments to the Author**

1. If the authors have adequately addressed your comments raised in a previous round of review and you feel that this manuscript is now acceptable for publication, you may indicate that here to bypass the “Comments to the Author” section, enter your conflict of interest statement in the “Confidential to Editor” section, and submit your "Accept" recommendation.

Reviewer #1: All comments have been addressed

2. Does this manuscript meet PLOS Global Public Health’s publication criteria? Is the manuscript technically sound, and do the data support the conclusions? The manuscript must describe methodologically and ethically rigorous research with conclusions that are appropriately drawn based on the data presented.

Reviewer #1: Yes

3. Has the statistical analysis been performed appropriately and rigorously?

Reviewer #1: Yes

4. Have the authors made all data underlying the findings in their manuscript fully available (please refer to the Data Availability Statement at the start of the manuscript PDF file)?

Reviewer #1: Yes

5. Is the manuscript presented in an intelligible fashion and written in standard English?

Reviewer #1: Yes

6. Review Comments to the Author

Reviewer #1: Thanks for addressing all the review feedback. This can now be published with some minor formatting edits.

7. PLOS authors have the option to publish the peer review history of their article (what does this mean?). If published, this will include your full peer review and any attached files.

**Do you want your identity to be public for this peer review?** For information about this choice, including consent withdrawal, please see our Privacy Policy.

Reviewer #1: **Yes: **Taufique Joarder
